

**A method for retrieving clouds with satellite infrared**
**radiances using the particle filter**
Dongmei Xu[1,2] , Thomas Auligné[2], Gaël Descombes[2], and Chris Synder[2]
[1]*Collaborative Innovation Center on Forecast and Evaluation of Meteorological*
*Disasters, Nanjing University of Information Science & Technology, Nanjing,* 210044,
*China*
[2]*National Center for Atmospheric Research, Boulder, Colorado* 80301, *USA*

9      (2016/7/2)

---

* **Corresponding Author**

Dr. Dongmei Xu

Nanjing University of Information Science & Technology, College of Atmospheric science,

Ningliu road, No. 219, Nanjing, 210044, China

E-mail: xdmjolly@sina.com





## Abstract

Ensemble-based techniques have been widely utilized in estimating uncertainties in
various problems of interest in geophysical applications. A new cloud retrieval
method is proposed based on the efficient Particle Filter (PF) in the framework of
Gridpoint Statistical Interpolation system (GSI). The PF cloud retrieval method is
compared with the Multivariate and Minimum Residual (MMR) method that was
previously established and verified. Cloud retrieval experiments involving a variety of
cloudy types are conducted with the PF and MMR methods respectively with
measurements of Infrared radiances on multi-sensors onboard both geostationary and
polar satellites. It is found that the retrieved cloud masks with both methods are
consistent with other independent cloud products. MMR is prone to producing
ambiguous small-fraction clouds, while PF detects clearer cloud signals, yielding
closer heights of cloud top and cloud base to other references. More collections of
small fraction particles are able to effectively estimate the semi-transparent high
clouds. It is found that radiances with high spectral resolutions contribute to
quantitative cloud top and cloud base retrievals. In addition, a different way of
resolving the filtering problem over each model grid is tested to better aggregate the
weights with all available sensors considered, which is proven to be less constrained
by the ordering of sensors. Compared to the MMR method, the PF method is overall
more computationally efficient, and the cost of the model grid-based PF method
scales more directly with the number of computing nodes.
Keywords: cloud retrieval methods, particle filter, GSI system, cloud height



## 1. Introduction


Modern polar orbiting and geostationary airborne instruments provide researchers

unprecedented opportunities for earth remote sensing with continuous flows and
almost complete spectral coverage of data. The primary cloud retrieval products from
satellites are cloud mask (CM), cloud height (CH), effective cloud fraction (CF), and
vertical structures of clouds with larger temporal and spatial scales. These cloud
retrievals provide an immense and valuable combination for better initializing
hydrometeors in numerical weather prediction (NWP) (Wu and Smith, 1992;Hu et al.,
2006;Bayler et al., 2000;Auligné et al., 2011), regulating the radiation budget for the
planet, and understanding the climate feedback mechanism (Rossow and Schiffer,
1991;Rossow et al., 1993;Brückner et al., 2014). Advanced cloud retrieval methods
are able to retrieve clouds with multispectral techniques (Menzel et al., 1983;Platnick
et al., 2003), among which the minimization methods usually directly utilize the
difference between the modeled clear sky and the observed cloudy Infrared (IR)
radiances (e. g., the minimum residual method, (Eyre and Menzel, 1989) the
Minimum Local Emissivity Variance method, (Huang et al., 2004); and the
Multivariate Minimum Residual method, (Auligné, 2014a). Specially, the
Multivariate Minimum Residual (MMR) method is retrieving three dimensional
multi-layer clouds by minimizing a cost function at each field-of-view (FOV)
(Auligné, 2014b;Xu et al., 2015). MMR has been proven to be reliable in retrieving
the quantitative three dimensional cloud fractions with Infrared radiances from



multiple infrared instruments. However, MMR has limitations in several aspects due
to its use of minimization for solution: 1) Part of the control variables accounting for
the cloud fraction for some certain levels are under-observed since the channels are
not sensitive to the existence of clouds for those heights. 2) When clouds at different
heights show opacities with the same spectral signal, MMR could lose the ability to
distinguish solutions involving clouds at those levels. 3) The computational cost for
the minimization procedure in MMR is rather considerable.

Ensemble-based techniques, that usually reside in short-term ensemble

forecasting (Berrocal et al., 2007), assembling existing model outputs (e. g., cloud
retrievals) from varying algorithms (Zhao et al., 2012), or ensemble Kalman filter
(EnKF) in various forms (Snyder and Zhang, 2003),   have been widely developed in
order to estimate the uncertainties of all kinds of problems in geophysical applications.
To better account for the non-linearity between the observed radiance and the retrieval
parameter, a novel prototype for detecting clouds and retrieving their vertical
extension inspired by the particle filter (Snyder et al., 2008;van Leeuwen, 2010;Shen
and Tang, 2015) technique and Bayesian theory (Karlsson et al., 2015) is proposed in
this study. As a competitive alternative for MMR, the PF retrieval method has same
critical inputs required and cloud retrieval products as in MMR. A brief description of
MMR and the new PF cloud retrieval algorithm are provided in the following section.
Section 3 describes the background model, the data assimilation system, the radiative
transfer models (RTMs), and the radiance observations applied in this study. Model
configurations are also illustrated in section 3. In section 4, the single test within one



FOV is conducted before the performance of PF method is assessed by comparing its
cloud retrievals with those from MMR and other operational cloud products. Section 4
also discusses the computational performance for the two methods. The conclusion
and anticipated future work are outlined in section 5.

## 2. Methodology


Essentially, the PF cloud retrieval scheme retrieves clouds with the same critical

inputs requested (i. e., clear sky radiance from the radiative transfer model and the
observed radiance) and the same cloud retrievals as outputs (i. e., effective three
dimensional cloud fractions) with the MMR method. Details of the schematic of the
MMR method can be referred in (Xu et al., 2015;Descombes et al., 2014). We use
$\boldsymbol{c} = c^1, c^2, ..., c^K$ to denote the array of vertical effective cloud fractions for K model
levels, and $c^0$ as the fraction of clear sky with $0 \le c^k \le 1, \ \forall k \in [0, K]$. In this study,
a cloud on one model level with a given fraction $c^k$ is assumed to block the
radiation from its lower model levels. The radiation originating from its lower levels
is assumed to contribute to the top of atmosphere radiance observed by the satellites
only with the residual fractions. The MMR method is an approach to retrieve cloud
fractions using the minimization technique. The residual of the modeled radiance and
the observation is normalized by the observed radiance, which results in the following
cost function:
$$J(c^0, \boldsymbol{c}) = \frac{1}{2} \sum_v \left[ \frac{R_v^{\text{cloud}} - R_v^{\text{obs}}}{R_v^{\text{obs}}} \right]^2, \qquad (1)$$





where $R_v^{\mathrm{cloud}}$ is the modeled cloudy radiance, and $R_v^{\mathrm{obs}}$ the observed radiance at
frequency $v$. This vertical cloud fraction $\boldsymbol{c} = c^1, c^2, ..., c^K$ and $c^0$ are control variables
for the cost function, where the simulated $R_v^{\mathrm{cloud}}$ is defined as

$$R_v^{\mathrm{cloud}}(c^0, c^1, c^2, ..., c^K) = c^0 R_v^0 + \sum_{k=1}^{K} c^k R_v^k, \qquad (2)$$

with $c^0 + \sum_{k=1}^{K} c^k = 1$ as the constraint. Here $R_v^k$ is the radiance calculated assuming
an overcast black cloud at the model level $k$ and $R_v^0$ the radiance calculated in the
clear sky.
While MMR retrieves the cloud fractions on each model vertical level by
minimizing a cost function, PF calculates posterior weights for each ensemble
member based on the observation likelihood given that member. In its simplest form,
PF works by initializing a collection of cloud profiles as particles and then estimating
the cloud distributions by averaging those particles with their corresponding weights.
Explicitly, each particle's weight is computed with the difference between the
modeled cloudy radiance from the particle and the observed radiance.
As the probabilities of the cloud distribution are fully presented by the initial
particles, of particular interest is to evaluate different particle initialization schemes in
the PF method. Two typical approaches for generating particles are firstly designed;
the first one is to generate the perturbed samples $P_b^i$ ( $\forall i \in [1, n]$ ) from the cloud
profile in the background denoted as $P_b(\boldsymbol{c} = c^0, c^1, ..., c^K)$ by inflating, deflating, and
moving the clouds with small magnitudes, where n is the sample size. The perturbed
cloud fractions are designated to replenish the ensemble by introducing the prior





informations of the cloud distributions from the background. Besides those perturbed
particles, to represent the existence of one-layer cloud on each model level with an
even chance, another diversity set of profiles $P^i_b$ ( $\forall i \in [1, K+1]$ ) are also initialized,
among which, $P^i$ stands for the profile with 100% cloud fraction on the model level $i$
($c^i$=100%) and 0% cloud on the rest levels. In particular, $P^0$ defines 100% clear ($c^0$=1).
It is also interesting to discretize the initial particles by setting the one-layer cloud
with the value of $c^i$ from 100% to 0% (e. g., 100%, 90%, 80%, …, 0% with 10% as
the interval) and further from 100% to 0% (e. g., 100%, 99%, 98%, 97%, …, 0% with
1% as the interval). For each particle $P^i_b$, its simulated cloudy radiance $R^{cloud}_{v,i}$ from the
model background can be obtained with Eq. (2). The weight $w^i$ for each particle $P^i_b$
thus is calculated by comparing the simulated $R^{cloud}_{v,i}$ and the observation $R^{obs}_v$ using
the exponential function as
$$w^i = e^{-\left(\frac{R^{obs}_v - R^{cloud}_{v,i}}{\sigma}\right)^2},$$
(3)

$\forall i \in [1, p]$. Here p is the particle size and $\sigma$ is the specified observation error. The
final analyzed $P_a$ is obtained by averaging the background particles $P^i_b$ with their
corresponding weight, as
$$P_a(c^0, c^1, c^2, ..., c^K) = \sum_{i=1}^{p} w^i P_b^i.$$
(4)

After updating all the particles, the final averaged cloud fractions $c^k_a$ are
normalized by
$$c^k_a = \frac{c^k}{\sum_{k=0}^{K} c^k},$$
(5)



where $\forall k \in [0, \mathrm{K}]$.

# 3. Data and model configurations

3.1 Data
The Advanced Infrared Sounder (AIRS), the Infrared Atmospheric Sounding
Interferometer (IASI), and the Cross-track Infrared Sounder (CrIs) are among the
most advanced hyperspectral infrared sounders and thus are applied for retrieving
clouds with hundreds of channels (Blumstein et al., 2004;Aumann et al., 2003;Xu et
al., 2013;Smith et al., 2015). The Radiance measurements from Moderate Resolution
Imaging Spectroradiometer (MODIS) onboard the Earth Observing System (EOS)
Terra or Aqua satellites are also well suited to extracting valuable cloud information
from the 36 spectral broadbands in the visible, near infrared and infrared regions at
high spatial resolution (1–5 km) (Ackerman, 1998). Apart from the IR radiances
from polar satellites, the Geostationary Operational Environmental Satellites (GOES)
Imager (Menzel and Purdom, 1994) provides a continuous stream of data over the
observing domain. In this study, GOES-13 (east) and GOES-15 (west) are utilized to
obtain cloud fractions over the continental United States (CONUS) domain. The
GOES Imager used in this study is a five-channel (one visible, four infrared)
imaging radiometer designed to sense radiant and solar reflected energy. The
instrument parameters for the sensors and the setups for channel selections can be
found in (Xu et al., 2015).



### 3.2 WRF, GSI and the radiative transfer model


The background fields are processed running the Weather Research and Forecast
(WRF) model (Skamarock et al., 2008). The MMR and PF cloud retrieval algorithms
are both implemented based on the gridpoint statistical interpolation data assimilation
system (GSI) (Wu et al., 2002; Kleist et al., 2009), which is a widely used data
assimilation system in operations and researches in NWP. GSI is capable of ingesting
a large variety of satellite radiance observations and has developed capabilities for
data thinning, quality control, and satellite radiance bias correction. The Community
Radiative Transfer Model (Liu and Weng, 2006; Han et al., 2006) was used as the
radiance forward operator for computing the clear-sky radiance and the radiance given
overcast clouds at each model level.
### 3.3 Model configurations
The WRF is configured with 415*325 horizontal grids at 15-km grid spacing, and
40 vertical levels up to 50 hPa within the single CONUS domain. The MMR and PF
cloud detection schemes search the cloud top using approximately 150 hPa as the
highest extent. Channels in the longwave region are utilized following the channel
selection scheme in (Xu et al., 2015). Generally, for each FOV, the retrieved cloud
fractions are extrapolated to its four neighboring model grid points. For polar satellite
pixels, the representative cloud fractions are extrapolated with an adaptive radius with
respect to their scan positions. The cloud detecting procedure for retrieving clouds is
conducted for each FOV from each individual sensor independently and sequentially.



## 4. Experiments and results


The PF experiments apply two groups of particles as mentioned in section 2,
among which the group-2 particles contains solely 100% one-layer clouds. To reveal
how the setup of the initial particles impacts the results, apart from the MMR and PF
experiments, we included another advanced experiment, denoted as APF. APF
requires more sampled particles including ranges of cloud fractions spanning from 0%
to 100% at the interval of 10%. An additional experiment "APFg2", similar to APF
but excluding the perturbed particles from the background in group-1 introduced in
section 2, was conducted to evaluate the added values from the group-one particles. In
this section, cloud retrieval experiments for several cases containing clouds of a
variety of types are conducted for comparison reason. The GOES imager retrieved
products from National Aeronautics and Space Administration (NASA-Langley cloud
and radiation products) are applied as a reference to validate the cloud retrieving
methods for the CONUS domain with a large and uniform coverage of cloud mask. In
addition, the retrieved cloud products were also compared to available CloudSat
(Stephens et al., 2002) and MODIS level-2 cloud products (Platnick et al., 2003)
archived by the CloudSat Data Processing Center in Colorado State and NASA
respectively.
4.1 Single test at one field of view
The PF cloud retrieving algorithm retrieves the cloud distributions by averaging
those initial particles with their weights. Before the real case experiments are carried

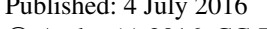



out over the whole domain, we conduct a single cloud retrieving test at one FOV to
understand what differences can be explained by the differences in the basic initial
particles. In Eq. (7), the observation error $\sigma$ can be set proportional to the
observation, equaling to $\dfrac{R_v^{obs}}{o\_f}$, where $o\_f$ is the prescribed ratio. Thus, the cloud
signals on each level $k$ are virtually determined by the extent of how close the $\dfrac{R_v^k}{R_v^{obs}}$
(and $\dfrac{R_v^0}{R_v^{obs}}$ for the clear part) gets to 1. An example of the ratio of the overcast
radiance and the observed radiance $\dfrac{R_v^k}{R_v^{obs}}$ for each model level is given in Fig. 1 of
GOES-Imager for the channel 5 (~13.00 $\mu m$). The clear sky radiance normalized by
the observed radiance $\dfrac{R_v^0}{R_v^{obs}}$ is also shown at the level 0 (Fig. 1). It is expected that
the overcast radiance from the RTM decrease with the rising of the altitude. The cloud
signal is strongest around level 5, where $R_v^k$ fits $R_v^{obs}$ most closely. The cloud
retrievals depend not only on the basic input profiles (i.e., the overcast radiance on
each level from RTM normalized by the observed radiance and the clear sky radiance
from RTM normalized by the observed radiance) and but also on the algorithm
applied for resolving the problem (e.g., MMR and PF in this study).






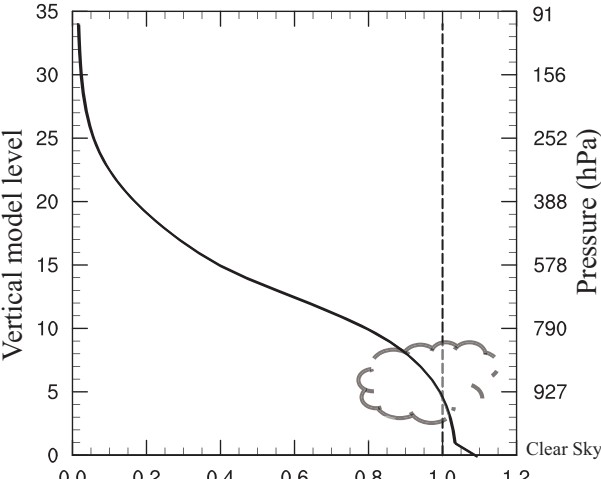

**Figure 1**. Ratio of the overcast radiances versus the observed radiance starting from the level 1.
The ratio of the clear sky radiance normalized by the observed radiance corresponds to the level 0
(see text for explanation) for GOES-Imager for the channel 5. The approximate pressures
corresponding to the model levels are also denoted.

To reveal the roles of various initial particles, Fig. 2a shows the weights for

different particles with specified value of cloud fractions (on the x-axis) on specified
model levels (on the y-axis) from 10% to 100% every 10% on the given FOV for
channel 5 of GOES-Imager for the case shown in Fig. 1. It seems that clouds can be
described by particles with both large fractions and small fractions. Low clouds are
easily estimated by one-layer cloud profile with large fractions (larger than 10%). The
particles with small-fraction high clouds gain some weights to retrieve high clouds.
The particle with the one-layer cloud on level 13 seems to gain least weight compared
to the others levels. The weights for the particles with cloud fractions from 0% to
100% at the interval of 1% are also presented in Fig. 2b. By including more
small-fraction one-layer clouds, the clouds around level 13 can be reproduced by the





group of refined particles with 1% as the interval for approximate 10% cloud fractions.
However, changing the level of the cloud for the fixed fraction (10%) does not seem
to change the outgoing radiance much, probably due to the channel's low weight
function peak (~750hPa).
The normalized $J_o$ for different levels with a specific cloud fraction from 0% to
100% every 10% are shown in the bottom panel of Fig. 2, with 10% and 1% as the
intervals in Fig. 2c and Fig. 2d respectively. From Fig. 2c, it is found that $J_o$ is
smallest around level-5 with 100% cloud fraction (denoted as 1 in legend) for the thin
black line, with respect to the fact that the overcast radiance fits the observed radiance
most closely for level-5 approximately. The grey line with 10% cloud fraction (0.1 in
the legend) corresponds to the existence of a weight peak on level 19 in Fig. 2a. In
addition, the gap between the grey line with 0.1 and the other lines from 0.2 to 1
explains why there's less continuity around level 13. Fig. 2d shows a similar pattern to
Fig. 2c, except with densely-distributed $J_o$ values around the level 13 from 0.1 to 1 in
the legend. Those contiguous black lines in Fig. 2d are associated with the set of
particles with cloud fractions from 10% to 100% at the interval of 1%.

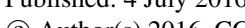

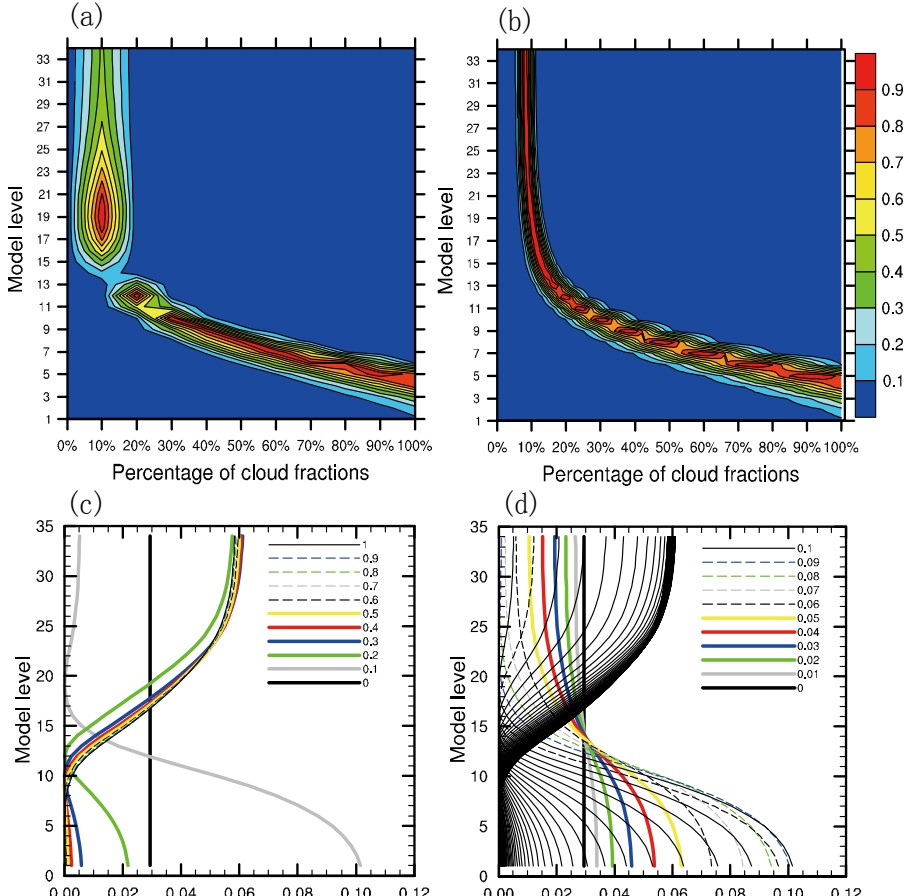


**Figure 2**. The weights for different particles with specified cloud fractions on the x-axis at one
chosen model level shown on the y-axis from 0% to 100% (a) at the interval of 10% and (b) at the
interval of 1%. The normalized $J_o$ (c) at the interval of 10% and (d) at the interval of 1%. In (d),
the normalized $J_o$ from 0.1 to 1 are all denoted as black lines.
4.2 Cloud profiles

The retrieval experiments for a real case are conducted at 1100 UTC 3 June 2012

when AIRS measurements and the CloudSat "2B-GEOPROF" products (Mace, 2004)
are available. The vertical cross sections of the cloud fraction field of a real case are



illustrated to further check how different collections of initial particles impact the
retrieved cloud profiles. The standard radar reflectivity profiles from the CloudSat are
shown in Fig. 3a as the validation source; Fig. 3b, Fig. 3c, and Fig. 3d show the cross
sections of the cloud fractions along the CloudSat orbit tracks from the MMR, PF and
APF experiments. The vertical structures of the clouds from MMR compare well with
the radar reflectivity from CloudSat by retrieving the high clouds around 47N° and
low clouds around 52N°. The PF experiment has difficulties in detecting the cloud
tops appropriately. PF tends to detect a large quantity of low clouds; by adding a set of
particles with small-fraction clouds in APF, higher clouds can be reproduced, which is
consistent with the implications from Fig. 2b and 2d. APF detects clear strong cloud
signals and removes the cloud fractions on near-surface levels around 36 N°
successfully. Since the existences of ground-layer radar reflectivity are likely
corresponding to the strong reflection from the underlying surface of the earth, the
height of cloud bases of MMR and PF are not compared in this sub-section. The
experiments with larger size of particles including 0% to 20% (at the interval of 1%)
plus 30% to 100% (at the interval of 10%) or of 0% to 100% (at the interval of 1%)
one-layer cloud profiles (introduced in section 2) yield similar results from APF but
are much more costly (not shown).





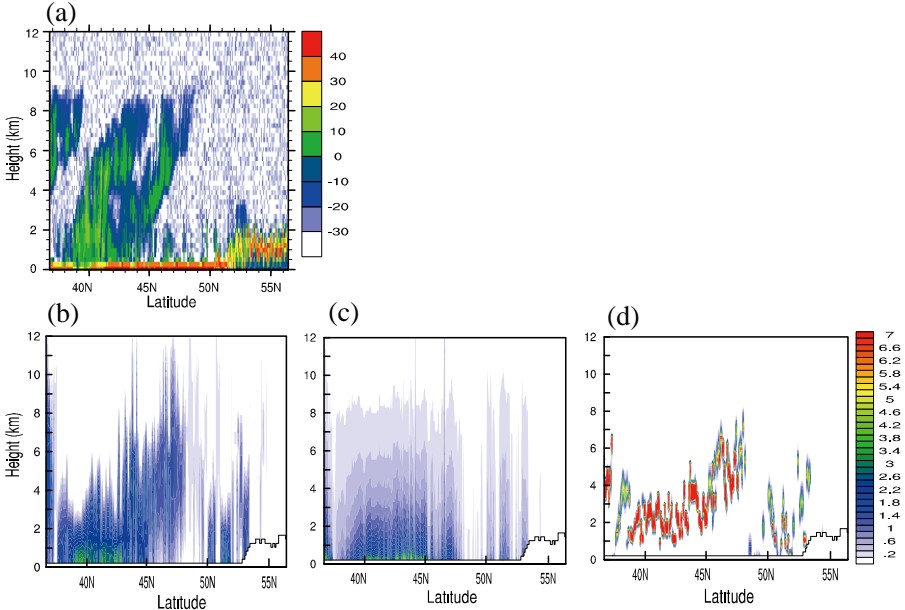

**Figure 3**. (a) The radar reflectivity (units: DBZ) cross sections from CloudSat, (b) the MMR retrieved cloud fractions (units: %) cross sections, (c) the PF retrieved cloud fractions, and (d) the APF retrieved cloud fractions valid at 1100 UTC 3 June 2012.

The vertical profiles of the averaged cloud fractions from MMR, PF, and APF are plotted in Fig. 4 at 1100 UTC 3 June 2012 with AIRS. Both MMR and PF experiments yield ambiguous cloud distributions, whereas APF retrieves much stronger cloud signals constrained between level-2 to level-20 (approximately from 950hPa to 400hPa). More clouds around level 10 are retrieved (approximately 750hPa) in MMR, while PF is prone to retrieving clouds near surface levels. Note that MMR retrieves much higher cloud tops and lower cloud bases compared to APF. The cloud base from PF is lowest; the cloud top from MMR and PF is comparable. Only the APF related methods will be further discussed in later sections owing to the missing of high clouds using PF.




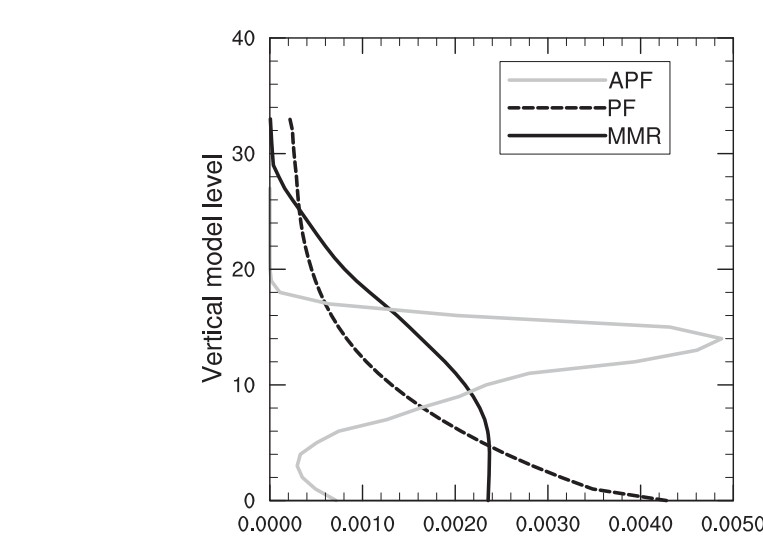


**Figure 4**. The mean cloud fraction on all model levels for the experiments MMR, PF, and APF

from AIRS valid at 1100 UTC 3 June 2012.
4.3 Cloud mask
Comparison experiments on real cases are further performed for over longer time
period from 0000 UTC 12 December 2013 to 0700 UTC 12 December 2013. The
cloud mask is marked as cloudy when there is a recognizable existence of cloud on
any level from MMR or PF retrievals. Both the NASA GOES Imager products and the
MMR-retrieved fields are interpolated to the same 0.1°×0.1° latitude–longitude grid
with 0 for clear and 1 for cloudy before the comparisons for verification. Fig. 5 shows
the *hits*, *false_alarms* and *misses* locations with the use of radiances from
GOES-Imager, MODIS, CrIS, AIRS, and IASI in the retrieval algorithms at 0700
UTC 12 December 2013. Note that, cloud mask retrievals from both the MMR and
APF hit the clear and cloudy events well in Fig. 5a and 5b. In most areas, the MMR





experiment overestimated the cloud mask with more false alarm events compared to
the APF experiment, since the MMR solution is an "overly smoothed" estimation of
the true vertical profile. It seems that the accuracy of cloud detection is lower for
areas with high altitude than under tropical conditions, indicating that the smaller
lapse rate in the atmosphere will lead radiance less sensitivity to clouds over polar
areas. Fig. 5c shows the cloud mask results from the APFg2 experiment without the
perturbed particles in group-1 introduced in section 2. There is no large discrepancy
between Fig. 5b and Fig. 5c, suggesting that the particles in group-2 that fully span
the possibility of the cloud distributions, are more determinant in retrieving the cloud
mask.

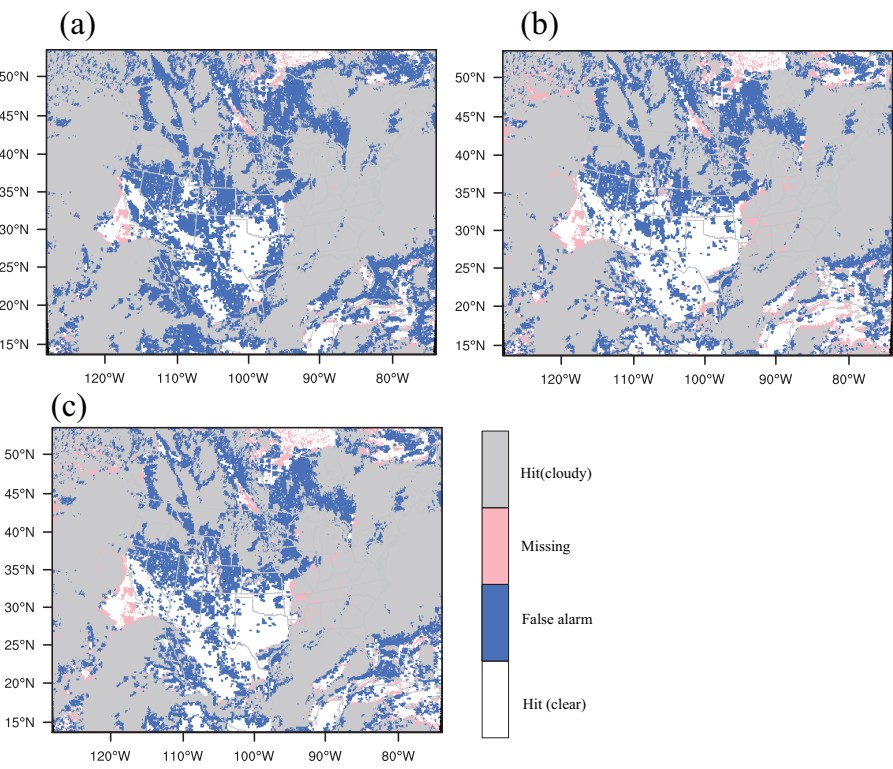




**Figure 5**. The false alarms, misses, and hits for clear and cloudy event locations with (a) the MMR
method, (b) the APF method, and (c) the APF method but without the group-1 particles (see text
for detailed explanations) valid at 0700 UTC 15 December 2013.
4.4 Cloud top and base pressure

The retrieved cloud top pressures (CTP) and cloud bottom pressures (CBP) from

this study along with the NASA GOES cloud products are illustrated in Fig. 6. The
CTPs from both methods are in good accordance with the NASA cloud products for
high clouds (from 100 hPa to 600 hPa) in Fig. 6a, 6c, and 6e. The retrieved cloud top
heights from MMR are overall higher than those from the NASA reference, especially
for lower clouds at approximately 750-1000 hPa (e. g., between longitude -100° and
-90°). On the other hand, the CTPs from APF are much closer to those in the
reference for both high and low clouds. APF overestimates the CBPs for some low
clouds (putting the clouds too low) in Fig. 6f; the overestimation of the CBP is even
more obvious from MMR in most regions in Fig. 6d.





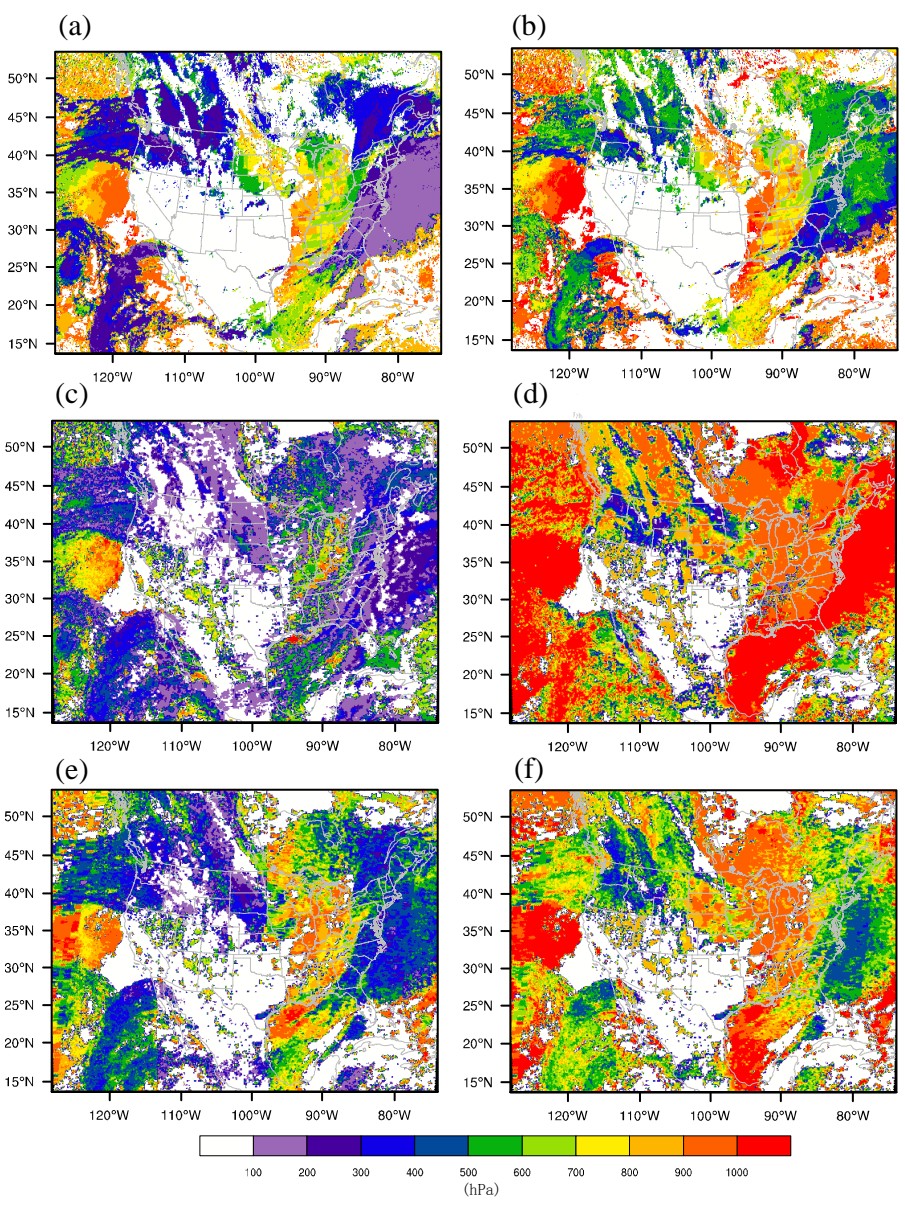

**Figure 6**. The cloud top pressure (left panels) from (a) the NASA GOES retrieval, (c) the MMR

method, (e) the APF method, and the cloud bottom pressure (right panels) from (b) the NASA

GOES retrieval, (d) the MMR method, (f) the APF method valid at 0700 UTC 15 December 2013.





The CTPs from NASA GOES cloud products for more hours (0300UTC,
0500UTC, 0700UTC) together with the independent CTP retrievals from MODIS
level-2 products (http://modis-atmos.gsfc.nasa.gov/MOD06_L2/) are plotted in Fig. 7.
Different sub-periods of the MODIS cloud retrieval products (e.g., Fig. 7b valid at
0320 UTC, Fig. 7c at 0325, and Fig. 7d at 0330 UTC) are chosen to approach the
valid times in Fig. 7a, Fig. 7e, and Fig. 7h respectively. The CTPs from both cloud
products agree well for both high and low clouds, confirming that NASA GOES cloud
products are overall reliable for verifying the cloud retrievals and MODIS level-2
products can also be applied for validations.



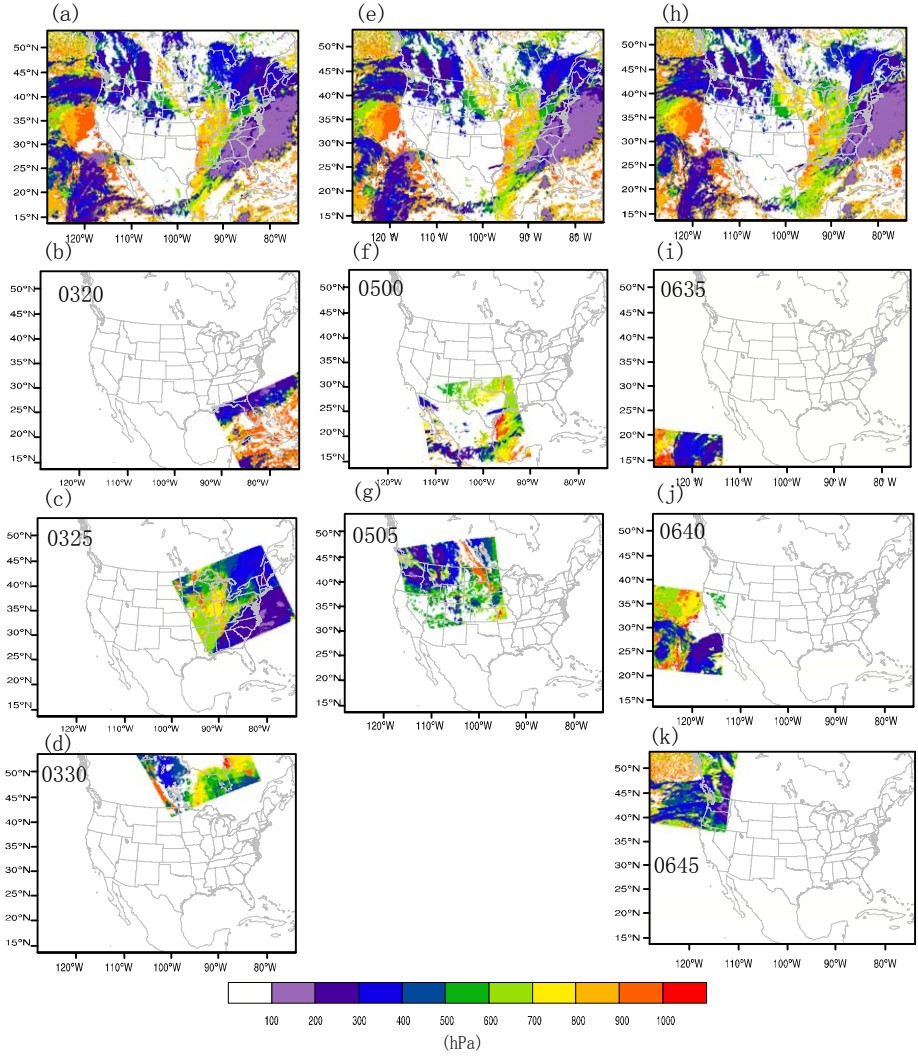

**Figure 7**. The cloud top pressure for (a) 0300UTC from the GOES NASA retrieval, (b) 0320UTC,

(c) 0325UTC, (d) 0330UTC from MODIS level-2 products; (e) 0500UTC from the GOES NASA

retrieval, (f) 0500UTC, (g) 0505UTC; (h) 0700UTC from the GOES NASA retrieval, (i)

0635UTC, (j) 0640UTC, and (k) 0645UTC from MODIS level-2 products.

Fig. 8 presents the correlation coefficients and biases of the CTP and CBP verified

against the NASA GOES and MODIS retrievals. The solid lines denote the results



regarding the CTP and CBP versus the NASA GOES products from 0000 UTC to
0700 UTC, while the dots describe the CTP results versus the cloud top retrievals in
NASA MODIS level-2 products at 0320UTC, 0325UTC, 0330UTC, 0500UTC,
0505UTC, 0635UTC, 0640UTC, and 0645UTC. Here the negative bias means that the
retrieved clouds are higher than the reference. Vice versa, the positive bias indicates
the clouds are put too low. We conducted another experiment "APFimg" that applies
solely GOES Imager data to check the added value from the high spectral resolution
radiances (such as, CrIS, AIRS, and IASI). In Fig. 8a, the correlations between the
retrievals from MMR and the NASA GOES retrievals are comparable with from APF
for most hours; APF gains overall higher correlations with the CTPs in the MODIS
retrievals. From the bias in Fig. 8b, it seems that the CTPs from MMR are
underestimated (putting the clouds too high) consistently against both retrievals from
GOES and MODIS. Fig. 8c shows that the correlations are weaker for MMR
compared to others all the time. In Fig. 8d, the positive CBP biases from MMR are
remarkable, while the CBP biases from APF are largely reduced. Generally, APFimg
degrades the CTP and CBP results consistently, suggesting that radiances with high
spectral resolutions are able to improve the vertical descriptions of cloud profiles. It is
found that the clouds retrieved with APFg2 are shrunken in terms of cloud depth with
notably lower cloud top and higher cloud base compared to APF, when excluding the
perturbed particles in the first group.





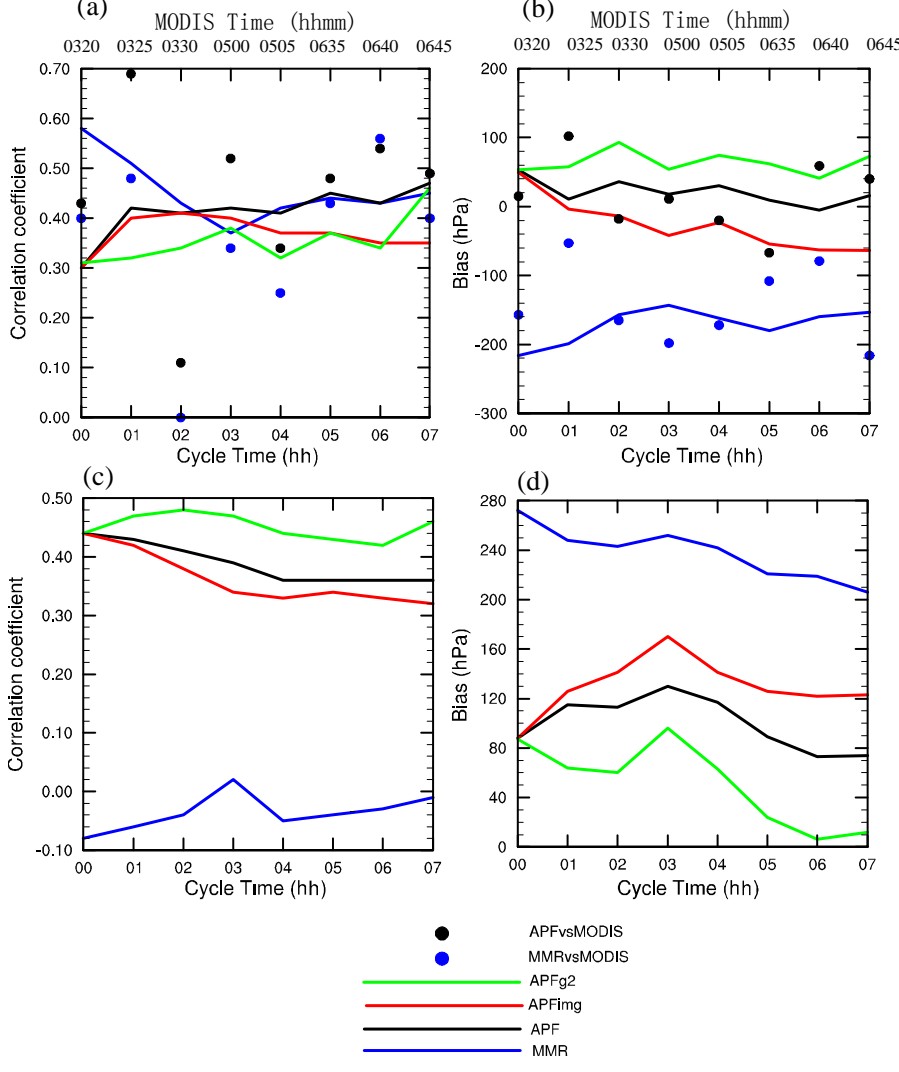


**Figure 8**. (a) Correlation coefficient, (b) bias for the cloud top pressure, (c) correlation coefficient,

and (d) bias for the cloud bottom pressure versus the NASA GOES retrievals from 0600 UTC 15

December 2013 to 0700 UTC 15 December 2013. Black and blue dots denote results versus the

MODIS level-2 cloud top pressure retrieval valid at 0320UTC, 0325UTC, 0330UTC, 0500UTC,

0505UTC, 0635UTC, 0640UTC, and 0645UTC. The valid times for the MODIS level-2 data are

shown on the top of the x-axis.



### 4.5 Computational issues

Fig. 9a represents the elapsed times for the MMR and APF experiments and the counts of radiance observations in use are shown in Fig.9b from 0000 UTC to 0700 UTC 12 December 2013. The profile of computing time in MMR is quite different from that in PF. The cost of MMR is dominated by the heavy minimization procedure, while APF is more associated with the processes of initializing particles and calculating weights for all the particles. The computing times were measured from cloud retrieving runs with 64 MPI-tasks on a single computing node in an IBM iDataPlex Cluster. The measured wall clock computing times show that generally MMR is computationally more expensive for most of the time than APF. It seems the wall clock times for MMR are generally proportional to the data amount used. While for the APF experiment, the wall clock time is mostly determined by the particles size and partly affected by the channel number, such as for 2013121202 and 2013121206, when the total counts of the hyperspectral sensors (IASI, CrIs, and AIRS) are large. The PF experiments using particles of one-layer cloud with 100% cloud fractions usually take less than 5 minutes for the same periods (not shown).



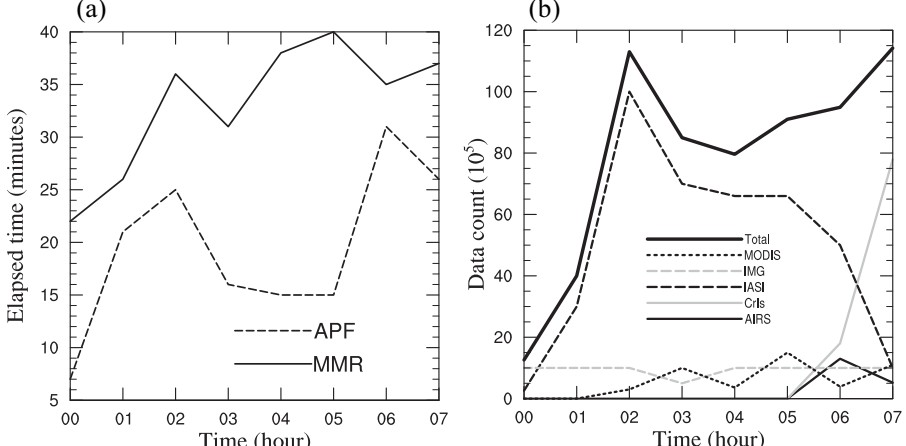

**Figure 9**. (a) The elapsed time and (b) the data count from 0000 UTC to 0700 UTC 15 December

2013.

4.6 Resolving the filtering problem on model grids

As explained in subsection 3.3, the filtering problem is resolved in the radiance

observational space at each FOV of each sensor independently and sequentially. For

each FOV, the retrieved cloud fractions are extrapolated to its neighboring model grid

points afterwards. We order the sensors in the cloud retrieving procedure as

GOES-Imager, MODIS, CrIS, AIRS, and IASI, aiming to optimize the vertical clouds

using sensors featured with sufficient spectral resolutions. As a consequence, the

retrievals from the last sensor determine the final output to the most extent, causing

the cloud retrievals highly subjective to the ordering of the sensors. On the other hand,

it means the information from other prior sensors will be more or less discarded. In

this section, a different way of resolving the filtering problem is preliminarily tested,

in which the weights for each particle are aggregated over all available sensors by



calling the forward radiative transfer model on neighbouring model grids.

Fig. 10 shows the clouds retrievals from the grid-based method. It is noted that

the grid-based scheme yields slightly worse results of CTP and neutral results of CBP
compared with those from the observation-based (FOV-based) scheme, indicating that
the hyperspectral sensors probably favor the retrieved CTP and CBP in the
FOV-based scheme, which are available for most of the time. It is worth pointing out
that the ordering of different sensors has nearly no effect on the final cloud retrievals,
when the weights of the particles are calculated in model space (not shown). The final
cloud retrieval is no longer overwritten by the retrieval from the last sensor but is a
total solution with all the sensors fairly considered, instead. The computational cost of
retrieving clouds in model space is comparable or slightly heavier than that in
observation space. The computational cost of the grid-based scheme scales with the
number of the computing nodes more directly, compared to that of the FOV-based
scheme.





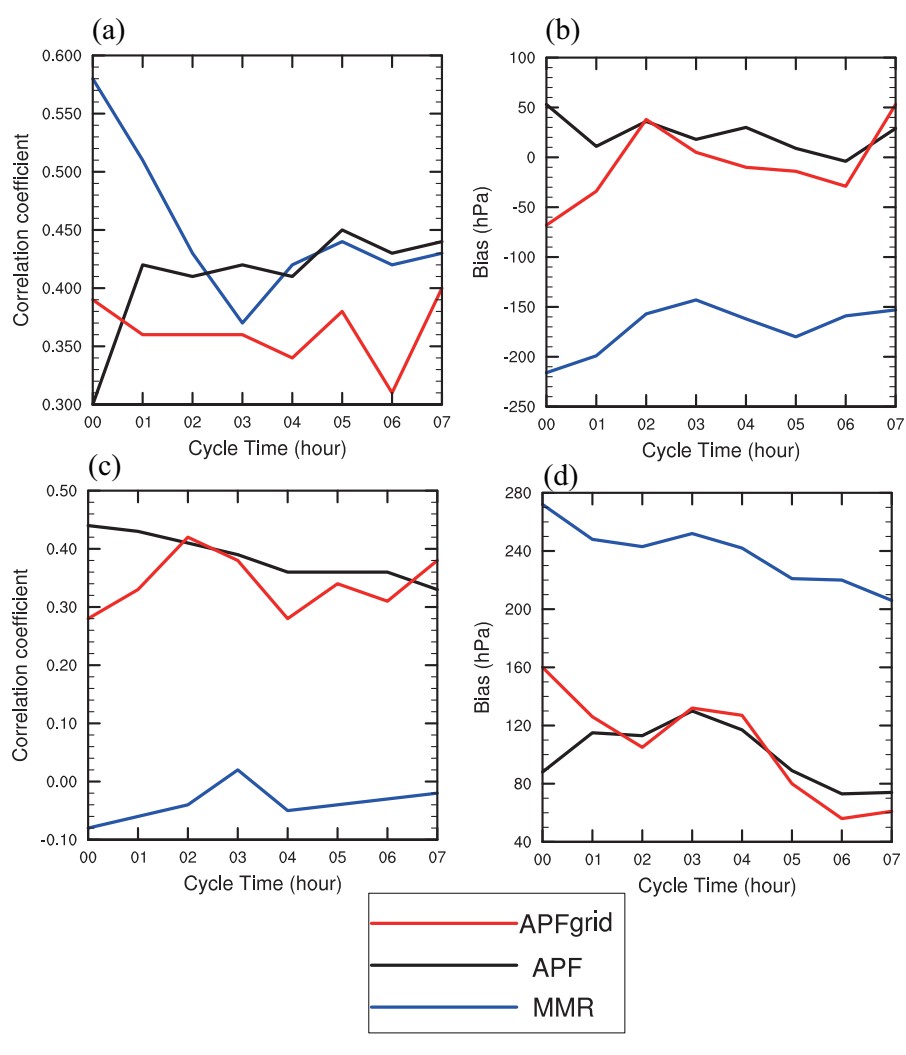




**Figure 10.** (a) Correlation coefficient, (b) bias for the cloud top pressure, (c) correlation

coefficient, and (d) bias for the cloud bottom pressure versus the NASA GOES retrievals from

0000 UTC to 0700 UTC 15 December 2013.

# 5. Discussion and conclusion

This study presents a new cloud retrieval method based on the particle filter (PF)

in the framework of GSI, as a competitive alternative to the MMR method. The

behaviors of different particle initializations are demonstrated on one single field of

view and the CONUS domain respectively. Comparisons between the PF and the

MMR method are conducted in terms of the features of cloud mask, cloud top, cloud

base, and the vertical distributions of clouds. It was found that the PF method

retrieves clear cloud signals while MMR is more ambiguous in detecting clouds. By

adding more small-fraction particles, high clouds can be better interpreted. From the

statistical results, it was found that MMR underestimates the cloud top pressures (put

the clouds top too high) and overestimates the cloud bottom pressures (put the clouds

top too low) as well. APF improves both the retrievals of cloud tops and cloud bases

remarkably, especially for the cloud bases. As expected, radiances with high spectral

resolutions contribute to quantitative cloud top and cloud base retrievals. In addition,

a different way of resolving the filtering problem over each model grid is tested to

aggregate the weights with all available sensors considered, which is proven to be less

constrained by the ordering of sensors. Last but not least, the PF method is overall



445 more computationally efficient; the cost of the model grid-based PF method scales

446 more directly with the number of the computing nodes.

447  In future work, validation studies using multispectral imagers on geostationary

448 satellites, spaceborne lidars (or radar), and surface site data will continue, and the

449 results will be used to update the retrieval algorithm. Maximizing the consistency in

450 the products across platforms and optimizing the synergistic use of multiple-source

451 radiances in the new algorithm are important aspects. To estimate the flow dependent

452 uncertainties in the cloud analysis and in the forecasts, the ensemble nowcasting with

453 three dimensional cloud fractions via the rapid-update cycling mode is also planned.

454 Finally, the use of cloud liquid water and ice mixing ratios retrieved from the cloud

455 fractions using multi-sensor radiances to pre-process the first guess in numerical

456 weather forecast is another promising application.

457 **Code and/or data availability**

458 The MMR cloud retrieval codes can be obtained freely from

459 (http://www2.mmm.ucar.edu/wrf/users/wrfda/). The other codes can be obtained by emails from

460 the authors.

461





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
