# Peer review of "A method for retrieving clouds with satellite infrared"

_Geoscientific Model Development, 2016_

## Referee Comment (RC1) · Anonymous Referee #1 · 29 Jul 2016

27/07/2016

Ensemble forecasts are produced as the routine products in many NWP operational centers. Using these products to estimate uncertainties is convenient and becomes popular.

This paper employed PF method, where the ensemble data is used to estimate cloud fraction, to retrieve cloud. The PF method provided improvements on the accuracies of cloud retrieval, i.e. cloud profile, cloud mask and cloud top, and also made the cost cheaper. This action is meaningful. Glad to see the study when satellite data is increasing in volume and plays the important role in providing extra information in NWP and data assimilation systems.

The extension of PF method to the cloud data retrieval is, I think, a good contribution to

the application of cloud products and a good addition to the literature. However, I see some deficiencies or ambiguous sentences in the paper that lead me to suggest major revisions before it is published. I see three major points that need to be corrected or explained.

1) The probabilities of the cloud distribution are presented by the initial particles. Thus, a particle initialization scheme is needed. Authors firstly generated the perturbations of cloud fractions by inflating, deflating and moving the clouds. My question is: (a) why did authors generate perturbations of cloud fractions when the cloud fractions were actually available among the ensemble members? Or say, why not use cloud fraction in ensemble members directly to generate the particles? If authors argue that the ensemble spread of cloud fractions in ensemble dataset is not large enough, it is reasonable but some statements should be stated here. (b) Did authors use any method and rule in this study to inflate, deflate and move the cloud? Some random perturbations might be deficient, and accounting for different perturbation methods could very well change some of the results in the basic PF experiment. I get the conclusion partly from Fig. 4, where the cloud fraction is obviously different between PF and APF and thus the different cloud retrievals are produced.

2) L174-175. "Generally, for each FOV, the retrieved cloud fractions are extrapolated to its four neighboring model grid points". What method is employed by authors to do the extrapolation from one cloud fraction to its neighboring grid points? Compared with the interpolation from background to FOV, which is a routine way to calculate the residual, is there any chance to make accuracy loss of radiance observations by the extrapolation? If so, how the loss of accuracy affects the weight in Eq. 3? I think authors need to tell the reader in more detail about this.

3) It is not a real question here. It is fine to use 150 hPa as the highest extent in this study. However, in reality, the tropopause could be higher than 150 hPa, e.g. an anvil cloud in a mature thunderstorm around tropopause at low latitude region. The fact can be also found out in Fig.4, where the cloud fraction around 150 hPa is not zero in

the experiment 'PF'. I do not ask authors to run extra experiments to estimate cloud fraction on all model levels because the cloud fraction is too small above 150 hPa and I consider this less important in this study. I just would like to say that we should not omit any extreme weather when we have the ability to resolve it.

That is a summary of my major concerns. The following are minor specific concerns generally relevant to specific portions of the text.

Line 13-16: If authors use the qualitative comments (L13-14) as the beginning, I suggest to add, say ' by using ensemble forecasts/products', behind PF in L15 to keep consistent to the L13-14. I don't think that all of readers are familiar with Particle Filter in which the ensemble concept is implicit when they read the abstract firstly, although the PF is introduced in section 2.

L48-50: Check parenthesis and comma, which do not match.

L98-101: Is c0 constant, if it is not the control variable ?

L101-102: Might state how Rvk is calculated briefly, e.g by forward CRTM operator with the inputs of temperature and humidity profiles in background.

L107: Suggest to note that the 'particles' correspond with 'ensemble members', i.e. one cloud profile as one of particles is derived from an ensemble member.

L131: If the observation error in Eq. 3 is specified in GSI, please state it.

L152: Only GOES-13 and -15 used in this study ? Does not match with Fig. 4.

L175: See major concern 2.

L189-196: Do authors implement bias correction for these satellite cloud products as reference ?

L202: Should be Eq. 3.

L216: Title of x-axis missed. Also check Fig. 2(c)(d).

L221 and L236: From Fig. 2a, I think the results are produced by using PF, because authors use these words "specified value of cloud fractions". However, the "normalized J0" is showed in Fig. 2c. It is confusing because MMR employs the cost function. If J0 is the residual in Eq. 3, please state it clearly.

L249: Fig. 2 could be separated into two figures, cloud fraction Fig. 2(a)(b) and normalized J0 Fig. 2(c)(d).

L293: I guess authors use AIRS as Robs to calculate residual, but need to re-write the word "from AIRS".

L372: Keep the units consistent. Check Fig. 9 and Fig. 10. Use (hour) or (hr), not (hh).

---

## Referee Comment (RC2) · Anonymous Referee #2 · 29 Jul 2016

Recommendation:

Minor revision

General comments:

The aim of the paper is to introduce a new retrieval cloud method, based on the particule filter approach. Since several very different configuration of cloud can lead to the same observed radiance, PF appears as nice tool for this problem. While similar use of the PF have been introduced in other domains (see comment 1 below), this is a new applications in this fields. The proposed method is compared with state of the art (MMR) where several particle generating technics have been considered. The results are well presented with an pedagogical situation to explore the potential of the method, and real cases. The benefit of the PF are a better retrieval at a lower cost compared

with the MMR. The manuscript can be improved to facilitate its reading following the comments, and minor revision are required.

Comments:

1) The bibliography on PF focuses on classical data assimilation consideration to estimate initial state. However, PF can also be used to parameter estimation or disaggregation which is similar to what introduced here, see eg Mechri et al. (2015). Hence you should clearly state the difference between the use of PF in classical DA and the present one, even if this relies on the same formalism, and improve the bibliography on this aspect.

2) Par 1, sec 2, l82: Precise the idea of cloud retrieval: this is implicit but for self consistency it is better to explain (generation of radiance from model, compared with observation, if they match then the cloud structure is found)..

3) l87: Precise the level associated with upper script $k$ (k=1 means near the surface .. or top atmosphere as encountered in NWP models – Fig. 1 explains it corresponds to the surface, but this should be written) ?

4) l87: "effective" is not clear, it should be better to explain as the fraction of top of cloud as seen from a sensor.

5) l88: Following the previous point 4), with the condition $0 \leq c^k \leq 1$, precise that $\sum_{k=0}^{K} c^k = 1$ at this place, with a label for this equation (the sum can be suppressed from l101).

6) l111: the definition of what is a particle is crucial since it use to be model state in classical dynamical system that is not the case here. Hence, you should precise explicitly that $P$ stands for the vector $c = (c^0, \cdots, c^K)$. In the notation, $P$ can be interpreted as a function $c^k$.. I think better to use $C = (c^0, \cdots, c^K)$ for the particle in place of the notation $P$ that could lead to confusion with the probability notation underlined with the particle filter approach. (see point 13 below)

7) l113: "typical" provide reference to previous work showing the method is known or suppress "typical".

8) l115: add an subscript $b$ to $c^k$ in $P_b$ as $c_b^k$

9) l115: "inflating, deflating, moving" should be illustrate using a regular 2D mesh, a simple figure would illustrates the fact that moving can suppress some fraction (a cloud becoming masked by another at upper level).

10) l111-126: the two approaches (l113) are not clearly separated, make two different paragraph one for each method (l114: the perturbation; l120 l123 the full/fractional one level top cloud)

11) l126: precise that for one-layer cloud at level $i$, the clear sky fraction is $c^0 = 1 - c^i$

12) l130: Eq.(3) means the comparison is done for one frequency.. what happens with other frequency (robustness, sensitivity) ? MMR relies on multiple frequency. At the opposite the PF seems to be used with only one. Please clarify this point / explain more precisely what is done.

13) l134: with the notation $C$, Eq.(4) becomes $C_a = \sum_i w_i C_b^i$ which is less confusing than with notation P.

14) l135: what is mean by updating ? (resampling strategy? analysis step?) I guess you mean analysis step for the particule filter, this should be clarified.

15) l135: precise that the average cloud fraction is no more normalised since the constraint (equation labelled from the above comments point 5) is not respected from the average Eq.(4) – average of state is no more a real state.

16) l202: Eq.(7) → Eq.(3)

17) l203: modify the notation for the prescribed ratio $o\_f$ is meaningless (use $r$, or something else, or explain why this notation is used).

[Figure]

18) l221-224: The particle used there corresponds to the groupe2 described previously, this should be reminded.

19) l224: Detail that the observation can be explained by different possible state and in particular as a fraction $c^i$ of one-cloud layer at a given level $i$ and a fraction of $c^0 = 1 - c^i$ of clearsky since $R_\nu^{cloud} = c^i R_\nu^i + (1 - c^i)R_\nu^0$ for levels $i$ upper than level $5$. Hence the theoretical one-layer cloud fraction is solution of $R_\nu^{obs} = c^i R_\nu^i + (1 - c^i)R_\nu^0$ that is by $c^i = \frac{R_\nu^0 - R_\nu^{obs}}{R_\nu^0 - R_\nu^i}$. No cloud can be present below level $5$ since this would implies an $R_\nu^{cloud}$ larger then the observation (or a $c^i$ larger than $100\%$). Provide a representation of the theoretical one-layer fraction so to introduce Fig2. This said, it is then easier to conclude that the weight in Fig2a 2b reproduce these possible situation with a maximum weight concentrated when the fraction is near the theoretical one given above.

20) l236: What is the normalized Jo ? I guess this should corresponds to the exponent in Eq.(3), but this is not introduced before. Provides the expression of Jo as a function of cloud fraction, it will be easier to understand what represents Fig. 2(c-d) $J_o(C^k) = o_f^2 \left( \frac{c^k R_\nu^k + (1 - c^k)R_\nu^0}{R_\nu^{obs}} - 1 \right)^2$, when $C^k = (0, \cdots, c^k, 0, \cdots, 0)$ with $c^k$ set to $0, 0.1, .., 1$ (c) and ... (d)

References:

Mechri, R.; Ottle, C.; Pannekoucke, O. Kallel, A. Genetic particle filter application to land surface temperature downscaling Journal Geophysical Research: Atmospheres, 2014, 119, 2131-2146.
* * *

---

## Author Comment (AC1) · 14 Aug 2016

Reply to Reviewer (1)'s comments on gmd-2016-150

We would like to thank the reviewer for the valuable comments and suggestions. Here are our responses to the reviewer's comments.

Comments to author:

Ensemble forecasts are produced as the routine products in many NWP operational centers. Using these products to estimate uncertainties is convenient and becomes popular.

This paper employed PF method, where the ensemble data is used to estimate cloud fraction, to retrieve cloud. The PF method provided improvements on the accuracies of cloud retrieval, i.e. cloud profile, cloud mask and cloud top, and also made the cost cheaper. This action is meaningful. Glad to see the study when satellite data is increasing in volume and plays the important role in providing extra information in NWP and data assimilation systems.

The extension of PF method to the cloud data retrieval is, I think, a good contribution to the application of cloud products and a good addition to the literature. However, I see some deficiencies or ambiguous sentences in the paper that lead me to suggest major revisions before it is published. I see three major points that need to be corrected or explained.

1) The probabilities of the cloud distribution are presented by the initial particles. Thus, a particle initialization scheme is needed. Authors firstly generated the perturbations of cloud fractions by inflating, deflating and moving the clouds. My question is: (a) why did authors generate perturbations of cloud fractions when the cloud fractions were actually available among the ensemble members? Or say, why not use cloud fraction in ensemble members directly to generate the particles? If authors argue that the ensemble spread of cloud fractions in ensemble dataset is not large enough, it is reasonable but some statements should be stated here. (b) Did authors use any method and rule in this study to inflate, deflate and move the cloud? Some random perturbations might be deficient, and accounting for different perturbation methods could very well change some of the results in the basic PF experiment. I get the conclusion partly from Fig.4, where the cloud fraction is obviously different between PF and APF and thus the different cloud retrievals are produced.
* * *
Reply: Basically, two groups of initial particles are used to retrieve the clouds in PF in this study. The first group includes particles that adding perturbations to clouds in the background in warm starts to further utilize prior information. The second group is to generate one-layer cloud on each model level with an even chance using different numbers of samples (e.g., with 100% interval, 10% interval, and 1% interval). There are two motivations of using the first group particles in addition to the particles in the second group. 1) Although cloud fractions can be actually retrieved with enough particles in group 2 with 1% interval, the computational cost is high, whereas the spread from particles with 10% or 100% intervals is not enough. 2) Since clouds move and evolve continuously in most cases, it may be more efficiently to generate a group of particles based on the warm background.

To make it clear, we added more explanations and statements as "The perturbed cloud fractions are designated to replenish the ensemble by introducing the prior information of the cloud distributions from the background and to increase the ensemble spread." and "the first one is to generate the perturbed samples $C_b^i$ ( $\forall i \in [1, n]$ ) from the cloud profile in the background denoted as $C_b = (c_b^0, c_b^1, ..., c_b^K)$ by inflating (deflating) the clouds with small magnitudes ( $C_b = \alpha \times C_b, \alpha = 50\%, 55\%, ..., 150\%$ ) and moving upward (downward) with $\delta z = +5, +4..., -1, ... -5$ as the vertical magnitude, where n is the sample size." in section 1.

2) L174-175. "Generally, for each FOV, the retrieved cloud fractions are extrapolated to its four neighboring model grid points". What method is employed by authors to do the extrapolation from one cloud fraction to its neighboring grid points? Compared with the interpolation from background to FOV, which is a routine way to calculate the residual, is there any chance to make accuracy loss of radiance observations by the extrapolation? If so, how the loss of accuracy affects the weight in Eq. 3? I think authors need to tell the reader in more detail about this.
* * *
Reply: Since the final retrieval clouds are on model grids, the retrieved cloud fractions within one FOV are essentially extrapolated to its four neighboring model grid points. Especially, for polar satellite pixels, the representative cloud fractions are extrapolated with an adaptive radius with respect to their scan positions. Since the clouds are retrieved FOV by FOV and the clouds on grids are referred immediately after one FOV is completed, there is no obvious accuracy loss of radiance observations using this conservative method. To make it clear, we added more related explanations and statements in the first paragraph in section 3.3.

3) It is not a real question here. It is fine to use 150 hPa as the highest extent in this study. However, in reality, the tropopause could be higher than 150 hPa, e.g. an anvil cloud in a mature thunderstorm around tropopause at low latitude region. The fact can be also found out in Fig.4, where the cloud fraction around 150 hPa is not zero in the experiment 'PF'. I do not ask authors to run extra experiments to estimate cloud fraction on all model levels because the cloud fraction is too small above 150 hPa and I consider this less important in this study. I just would like to say that we should not omit any extreme weather when we have the ability to resolve it.
* * *
Reply: We agree that in reality the tropopause could be higher than 150 hPa occasionally with very small probability. In fig. 4 (using the same y-aixs with fig. 2), all the experiments are already conducted using150 hPa as the highest extent. The reason that the averaged cloud fraction (cf) around 150 hPa is not exactly zero is that the cf is strictly for each model level. The pressure levels on the right y-axis are estimated roughly using the domain averaged pressure. We agree that any extreme weather should not be omitted and thus we added more explanations and statements as

"The MMR and PF cloud detection schemes search the cloud top using approximately 150 hPa as the highest extent for most cloudy cases. Other clouds higher 150 hPa, e.g. an anvil cloud in a mature thunderstorm around tropopause at low latitude region will also be explored in future studies." in section 3.3 and "Increasing the highest extent cloudy cases will be included in future studies. " in the future plan.

That is a summary of my major concerns. The following are minor specific concerns generally relevant to specific portions of the text.

Line 13-16: If authors use the qualitative comments (L13-14) as the beginning, I suggest to add, say 'by using ensemble forecasts/products', behind PF in L15 to keep consistent to the L13-14. I don't think that all of readers are familiar with Particle Filter in which the ensemble concept is implicit when they read the abstract firstly, although the PF is introduced in section 2.
* * *
Reply: A new cloud retrieval method is proposed based on the efficient Particle Filter (PF) by using ensembles of cloud information in the framework of Gridpoint Statistical Interpolation system (GSI).

L48-50: Check parenthesis and comma, which do not match.
* * *
Reply: Corrected.

L98-101: Is c0 constant, if it is not the control variable?
* * *
Reply: $c^0$ is one of the control variables in the cost function of MMR, instead of a constant. We add more explanations as " The residual of the modeled radiance and the observation is normalized by the observed radiance, which results in the following cost function, using $c^k$, $\forall k \in [0, \mathrm{K}]$ as the control variables"

L101-102: Might state how $R_v^k$ is calculated briefly, e.g by forward CRTM operator with the inputs of temperature and humidity profiles in background.
* * *
Reply: Accepted. We add statements as "Both $R_v^k$ and $R_v^0$ are calculated using a forward radiative transfer model with model profiles of temperature and moisture as inputs." in the first paragraph of section 2.

L107: Suggest to note that the 'particles' correspond with 'ensemble members', i.e. one cloud profile as one of particles is derived from an ensemble member.
* * *
Reply: Accepted. We add "Explicitly, the definition of particles corresponds with ensemble members, i.e. one cloud profile as one of particles is corresponding to an ensemble member." in the fourth paragraph of section 2 to state the definition of particles clearly.

L131: If the observation error in Eq. 3 is specified in GSI, please state it.
* * *
Reply: Here p is the particle size and $\sigma$ is the specified observation error, which can be referred in the first paragraph in section 4.1.

L152: Only GOES-13 and -15 used in this study? Does not match with Fig. 4.
* * *
Reply: The sentence is revised as "In this study, GOES-13 (east) and GOES-15 (west) are also utilized to obtain cloud fractions over the continental United States (CONUS) domain."

L175: See major concern 2.
* * *
Reply: See detailed reply to the above second major points of comments.

L189-196: Do authors implement bias correction for these satellite cloud products as reference?
* * *
Reply: No. That's why we utilized multiple cloud products for comprehensively comparisons.

L202: Should be Eq. 3.
* * *
Reply: Corrected. Since we added two new equations in ahead of Eq. (3), Eq. (3) is labelled as Eq. (5) in the revised manuscript.

L216: Title of x-axis missed. Also check Fig. 2(c)(d).
* * *
Reply: Done.

L221 and L236: From Fig. 2a, I think the results are produced by using PF, because authors use these words "specified value of cloud fractions". However, the "normalized J0" is showed in Fig. 2c. It is confusing because MMR employs the cost function. If J0 is the residual in Eq. 3, please state it clearly.
* * *
Reply: Yes. Fig. 2ab and Fig. 2cd are all using PF. We add more explanations in the sixth paragraph in section 2 as "A cost function Jo is defined for each particle to measure how the particle fit the observation as,

$$J_o = (\frac{R_v^{obs} - R_{v,i}^{cloud}}{\sigma})^2 \qquad (4)$$

" to state it clearly.

L249: Fig. 2 could be separated into two figures, cloud fraction Fig. 2(a) (b) and normalized J0 Fig. 2(c)(d).
* * *
Reply: Done.

L293: I guess authors use AIRS as Robs to calculate residual, but need to re-write the word "from AIRS".
* * *
Reply: Corrected. We re-write as "with AIRS observations".

L372: Keep the units consistent. Check Fig. 9 and Fig. 10. Use (hour) or (hr), not (hh).
* * *
Reply: Corrected.

[revised manuscript text omitted]

---

## Author Comment (AC2) · 15 Aug 2016

Reply to Reviewer (2)'s comments on gmd-2016-150

We would like to thank the reviewer for careful and thorough reading of this manuscript and for the constructive suggestions. Here are our responses to the reviewer's comments.

Comments to author:

General comments:

The aim of the paper is to introduce a new retrieval cloud method, based on the particle filter approach. Since several very different configuration of cloud can lead to the same observed radiance, PF appears as nice tool for this problem. While similar use of the PF have been introduced in other domains (see comment 1 below), this is a new applications in this fields. The proposed method is compared with state of the art (MMR) where several particle generating techniques have been considered. The results are well presented with an pedagogical situation to explore the potential of the method, and real cases. The benefit of the PF are a better retrieval at a lower cost compared with the MMR. The manuscript can be improved to facilitate its reading following the comments, and minor revision are required.

Comments:

1) The bibliography on PF focuses on classical data assimilation consideration to estimate initial state. However, PF can also be used to parameter estimation or disaggregation which is similar to what introduced here, see eg Mechri et al. (2015). Hence you should clearly state the difference between the use of PF in classical DA and the present one, even if this relies on the same formalism, and improve the bibliography on this aspect.
* * *
Reply: We reorganized the methodology part and added statements as "Particle filter (PF) approach is one of the nonlinear filters for data assimilation procedures to best estimate the initial state of a system or its parameters $x_t$, which describes the time evolution of the full probability density function $p(x_t)$ conditioned by the dynamics and the observations. Similar to (Mechri et al., 2014), the bibliography on PF focuses on estimating the parameters, which are the cloud fractions $c^k$ in Eq. (3), in this study." in paragraph 3 in section 2.

2) Par 1, sec 2, L82: Precise the idea of cloud retrieval: this is implicit but for self consistency it is better to explain (generation of radiance from model, compared with observation, if they match then the cloud structure is found).
* * *
Reply: Agreed. More statements are added as "Both cloud retrieval schemes consist of finding cloud fractions that allow best fit between the cloudy radiance from model and the observation." in the first paragraph in section 2.

3) L87: Precise the level associated with upper script $k$ ($k$=1 means near the surface .. or top atmosphere as encountered in NWP models – Fig. 1 explains it corresponds to the surface, but this should be written) ?
* * *
Reply: Accepted. In the revised manuscript, "We use $c^1, c^2, ..., c^K$ to denote the array of vertical effective cloud fractions for K model levels ( $c^1$ for the surface and $c^K$ for the model top) and $c^0$ as the fraction of clear sky with $0 \leq c^k \leq 1, \ \forall k \in [0, K]$ . " in section 2.

4) L87: "effective" is not clear, it should be better to explain as the fraction of top of cloud as seen from a sensor.
* * *
Reply: Accepted. We revised the statements as "Essentially, the PF cloud retrieval scheme retrieves clouds with the same critical inputs requested (i. e., clear sky radiance from the radiative transfer model and the observed radiance) and the same cloud retrievals as outputs (i. e., three dimensional cloud fractions, which is defined as the fraction of top of cloud as seen from a sensor) with the MMR method." in place of effective three dimensional cloud fractions).

5) L88: Following the previous point 4), with the condition $0 \le c^k \le 1$, precise that

$$\sum_{k=0}^{K} c^k = 1$$ at this place, with a label for this equation (the sum can be suppressed from L101).
* * *
Reply: Agreed. We labelled the equation and suppressed the sum from L101.

6) L111: the definition of what is a particle is crucial since it use to be model state in classical dynamical system that is not the case here. Hence, you should precise explicitly that P stands for the vector $\boldsymbol{c} = (c^0, c^1, ..., c^K)$. In the notation, P can be interpreted as a function ck.. I think better to use $\boldsymbol{C} = (c^0, c^1, ..., c^K)$ for the particle in place of the notation P that could lead to confusion with the probability notation underlined with the particle filter approach. (see point 13 below)
* * *
Reply: Accepted. We adopted the reviewer's idea that using $\boldsymbol{C} = (c^0, c^1, ..., c^K)$ to interpret the particle, which makes the notations more clear.

7) L113: "typical" provide reference to previous work showing the method is known or suppress "typical".
* * *
Reply: Agreed. We deleted "typical" in the sentence.

8) L115: add an subscript b to $c^k$ in $P_b$ as $c_b^k$
* * *
Reply: Done.

9) L115: "inflating, deflating, moving" should be illustrate using a regular 2D mesh, a simple figure would illustrates the fact that moving can suppress some fraction (a cloud becoming masked by another at upper level).
* * *
Reply: Done. The first one is to generate the perturbed samples $C_b^i$ ( $\forall i \in [1, n]$ ) from the cloud profile in the background denoted as $C_b = (c_b^0, c_b^1, ..., c_b^K)$ by inflating (deflating) the clouds with small magnitudes ( $C_b = \alpha \times C_b, \alpha = 50\%, 55\%, ..., 150\%$ ) and moving upward (downward) with $\delta z = +5, +4..., -1, ... -5$ as the vertical magnitude, where n is the sample size.

10) L111-126: the two approaches (L113) are not clearly separated, make two different paragraph one for each method (L114: the perturbation; L120 L123 the full/fractional one level top cloud)
* * *
Reply: Accepted.

11) L126: precise that for one-layer cloud at level i, the clear sky fraction is $c^0 = 1 - c^i$
* * *
Reply: Accepted.

12) L130: Eq.(3) means the comparison is done for one frequency.. what happens with other frequency (robustness, sensitivity) ? MMR relies on multiple frequency. At the opposite the PF seems to be used with only one. Please clarify this point / explain more precisely what is done.
* * *
Reply: PF also is conducted based on multiple frequency. We revised the manuscript as "The weight $w^i$ for each particle $C_b^i$ thus is calculated by comparing the simulated $R_{v,i}^{cloud}$ and the observation $R_v^{obs}$ using the exponential function by accumulating the Jo for multiple frequency as

$$w^i = e^{-\sum_v (\frac{R_v^{obs} - R_{v,i}^{cloud}}{\sigma})^2},$$

(5)

$\forall i \in [1, p]$." in sixth paragraph in section 2.

13) L134: with the notation C, Eq.(4) becomes $C_a = \sum_{i=1}^{p} w^i P_b^i$ which is less confusing than with notation P.
* * *
Reply: Accepted.

14) L135: what is mean by updating ? (resampling strategy? analysis step?) I guess you mean analysis step for the particule filter, this should be clarified.
* * *
Reply: Corrected. The revised sentence is "After the analysis step for the particle filter, the final averaged cloud fractions..."

15) L135: precise that the average cloud fraction is no more normalised since the constraint (equation labelled from the above comments point 5) is not respected from the average Eq.(4) – average of state is no more a real state.
* * *
Reply: Agreed. We added statements as "In Eq. (6), the constraint referred in Eq. (1) is not respected. Thus, after the analysis step for the particle filter, the final averaged cloud fractions $c_a^k$ are normalized by..."

16) L202: Eq.(7) --->Eq.(3)
* * *
Reply: Corrected. Since we added two new equations in ahead of Eq. (3), Eq. (3) is labelled as Eq. (5) in the revised manuscript.

17) L203: modify the notation for the prescribed ratio o_f is meaningless (use r, or something else, or explain why this notation is used).
* * *
Reply: Agreed.

We re-wrote the sentence as "In Eq. (3), the observation error $\sigma$ can be set proportional to the observation, equaling to $\dfrac{R_v^{obs}}{r}$, where $r$ is the prescribed ratio."

in the revised manuscript.

18) L221-224: The particle used there corresponds to the groupe2 described previously, this should be reminded.
* * *
Reply: Agreed.

In second paragraph of section 4.1., we added explanations of particles as "To reveal the roles of various initial particles, Fig. 2a shows the weights for different particles of one-layer cloud in group 2 described in section 2 with specified value of cloud fractions (on the x-axis) on specified model levels (on the y-axis) from 10% to 100% every 10% on the given FOV for channel 5 of GOES-Imager for the case shown in Fig. 1."

19) L224: Detail that the observation can be explained by different possible state and in particular as a fraction $c^i$ of one-cloud layer at a given level i and a fraction of $c^0 = 1 - c^i$ of clear sky since $R_v^{cloud} = c^i R_v^i + (1 - c^i) R_v^0$ for levels i upper than level

5. Hence the theoretical one-layer cloud fraction is the solution of

$R_v^{obs} = c^i R_v^i + (1-c^i)R_v^0$ that is by $c^i = \dfrac{R_v^0 - R_v^{obs}}{R_v^0 - R_v^i}$ . No cloud can be present below level 5 since this would imply an $R_v^{cloud}$ larger then the observation (or a $c^i$ larger than 100%). Provide a representation of the theoretical one-layer fraction so to introduce Fig2. This said, it is then easier to conclude that the weight in Fig2a 2b reproduce these possible situation with a maximum weight concentrated when the fraction is near the theoretical one given above.
* * *
Reply: Accepted. We add theoretical representation in the second paragraph in section 4.1 as "With a fraction $c^k$ of one-cloud layer at a given level k and a fraction of $c^0 = 1 - c^k$ of clear sky, the simulated cloudy radiance can be denoted as

$R_v^{cloud} = c^k R_v^k + (1-c^k)R_v^0$ . Hence the theoretical one-layer cloud fraction is solved as

$c^k = \dfrac{R_v^0 - R_v^{obs}}{R_v^0 - R_v^k}$ by fitting $R_v^{cloud}$ to $R_v^0$ . As expected, for one-layer cloud with full fraction, $c^5$ equals to 100% . Since with the concept that $R_v^k > R_v^{k+1}$ , no cloud can be present below level 5 since this would implies a $R_v^{cloud}$ larger than the observation (or a $c^i$ larger than 100%)."

20) L236: What is the normalized Jo ? I guess this should corresponds to the
exponent
in Eq.(3), but this is not introduced before. Provides the expression of Jo as a
function
of cloud fraction, it will be easier to understand what represents Fig. 2(c-d)
, when $C^k = (0,...,c^k,0,...0)$ with $c^K$ set to 0, 0.1,...1 (c) and ...(d)
* * *

[revised manuscript text omitted]

Administrator 2016-08-10 22:03

anna 2016-08-06 19:54

anna 2016-08-06 19:54

anna 2016-08-06 20:00

$c^k$ in Eq. (3), in this study. While MMR retrieves the cloud fractions on each model vertical level by minimizing a cost function, PF calculates posterior weights for each ensemble member based on the observation likelihood given that member. In its simplest form, PF works by initializing a collection of cloud profiles as particles and then estimating the cloud distributions by averaging those particles with their corresponding weights. Explicitly, each particle's weight is computed with the difference between the modeled cloudy radiance from the particle and the observed radiance.

As the probabilities of the cloud distribution are fully presented by the initial particles, of particular interest is to evaluate different particle initialization schemes in the PF method. Explicitly, the definition of particles corresponds with ensemble members, i.e. one cloud profile as one of particles is corresponding to an ensemble member.

Two approaches for generating particles are firstly designed; the first one is to generate the perturbed samples $C_b^i$ ( $\forall i \in [1, n]$ ) from the cloud profile in the background denoted as $C_b = (c_b^0, c_b^1, ...., c_b^K)$ by inflating (deflating) the clouds with small magnitudes ( $C_b = \alpha \times C_b, \alpha = 50\%, 55\%, ...., 150\%$ ) and moving upward (downward) with $\delta z = +5, +4..., -1, ... -5$ as the vertical magnitude, where n is the sample size. The perturbed cloud fractions are designated to replenish the ensemble by introducing the prior information of the cloud distributions from the background and to increase the ensemble spread.

Besides those perturbed particles, to represent the existence of one-layer cloud on each model level with an even chance, another diversity set of profiles $C_b^i$

( $\forall i \in [0, K]$ ) are also initialized, among which, $C_b^i$ stands for the profile with 100%

cloud fraction on the model level $i$ ($c^i$=100%) and 0% cloud on the rest levels. In particular, $C_b^0$ defines 100% clear ($c^0$=1). It is also interesting to discretize the initial particles by setting the one-layer cloud with the value of $c^i$ from 100% to 0% (e. g.,

100%, 90%, 80%, …, 0% with 10% as the interval) and further from 100% to 0% (e.

g., 100%, 99%, 98%, 97%, …, 0% with 1% as the interval). In this cases, $c^0$=1-$c^i$ . For each particle $C_b^i$, its simulated cloudy radiance $R_{v,i}^{\mathrm{cloud}}$ from the model background can be obtained with Eq. (2).

A cost function $J_o$ is defined for each particle to measure how the particle fit the observation as,

$$J_o = (\frac{R_v^{\mathrm{obs}} - R_{v,i}^{\mathrm{cloud}}}{\sigma})^2. \tag{4}$$

The weight $w^i$ for each particle $C_b^i$ thus is calculated by comparing the simulated

$R_{v,i}^{\mathrm{cloud}}$ and the observation $R_v^{\mathrm{obs}}$ using the exponential function by accumulating the

$J_o$ for multiple frequency as

$$w^i = e^{-\sum_v (\frac{R_v^{\mathrm{obs}} - R_{v,i}^{\mathrm{cloud}}}{\sigma})^2}, \tag{5}$$

$\forall i \in [1, p]$ . Here p is the particle size and $\sigma$ is the specified observation error, which can be referred in the first paragraph in section 4.1. The final analyzed $C_a$ is obtained by averaging the background particles $C_b^i$ with their corresponding weight, as anna 2016-07-30 20:25

[revised manuscript text omitted]

---

## Referee Comment (RC3) · Anonymous Referee #1 · 19 Aug 2016

Comments to author:

The authors have done some work to appropriately address my remarks and provide answers. The relevant discussion was also included at several places throughout the manuscript. I therefore recommend the manuscript be accepted for publication once the authors make the following typos that should still be considered.

L15: May delete 'efficient'.

L39: Consider using 'remote sensing of earth' instead of 'earth remote sensing'.

L51: May use [ ] instead of the outer ().

L66: May use 'various' or other words instead of 'all kinds of' ?

L118: Similar to the study (or implementation ) in Mechri et al……

L125: May delete 'Explicitly'.

L136: The equation never happens in mathematic unless $\alpha$ is 1. May use another letter for the left $C_b$.

L150: Eq.2 –> Eq. 3

L154: Move the definition of $\sigma$ at L158 to this line.

L158: Is n at L137 the same as p at L158 ? If so, why not use the same letter, n or p ?

L255: The sentence is too long. Rewrite it.

L297: Fig. 2 looks fine. However, cloud fraction and Normalized Jo are not the same thing. It may be better to consider Normalized Jo as Fig. 3.

---

## Author Comment (AC3) · 23 Aug 2016

Reply to Reviewer (1)'s comments on gmd-2016-150

We would like to thank the reviewer for further reading through this manuscript and the comments to improve this manuscript. Here are our responses.

Comments to author:

The authors have done some work to appropriately address my remarks and provide answers. The relevant discussion was also included at several places throughout the manuscript. I therefore recommend the manuscript be accepted for publication once the authors make the following typos that should still be considered.

L15: May delete 'efficient'.
* * *
Reply: Agreed.

L39: Consider using 'remote sensing of earth' instead of 'earth remote sensing'.
* * *
Reply: Accepted.

L51: May use [ ] instead of the outer ().
* * *
Reply: Agreed.

L66: May use 'various' or other words instead of 'all kinds of' ?
* * *
Reply: Agreed. The revised sentence is "…in diversified forms (Snyder and Zhang, 2003), have been widely developed in order to estimate the uncertainties of various problems in geophysical applications."

L118: Similar to the study (or implementation ) in Mechri et al······
* * *
Reply: Corrected. The sentence is revised as "Similar to the study in Mechri et al······ ".

L125: May delete 'Explicitly'.
* * *
Reply: Accepted.

L136: The equation never happens in mathematic unless $\alpha$ is 1. May use another letter for the left Cb.
* * *
Reply: Corrected. We modified the equation as $C_b^i = \alpha \times C_b, \alpha = 50\%, 55\%, ..., 150\%$.

L150: Eq.2 –> Eq. 3
* * *
Reply: Done.

L154: Move the definition of $\sigma$ at L158 to this line.
* * *
Reply: Accepted.

L158: Is n at L137 the same as p at L158 ? If so, why not use the same letter, n or p ?
* * *
Reply: Corrected. We used n consistently in the revised manuscript.

L255: The sentence is too long. Rewrite it.
* * *
Reply: We re-organized the sentence as two separate sentences.

L297: Fig. 2 looks fine. However, cloud fraction and Normalized Jo are not the same thing. It may be better to consider Normalized Jo as Fig. 3.
* * *
Reply: Agreed. We modified Fig. 2c and Fig. 2d to be Fig. 3a and Fig. 3b. The following Figure captions are also corrected accordingly.

[revised manuscript text omitted]